# High-Throughput Sequencing Reveals Differential Begomovirus Species Diversity in Non-Cultivated Plants in Northern-Pacific Mexico

**DOI:** 10.3390/v11070594

**Published:** 2019-06-29

**Authors:** Edgar Antonio Rodríguez-Negrete, Juan José Morales-Aguilar, Gustavo Domínguez-Duran, Gadiela Torres-Devora, Erika Camacho-Beltrán, Norma Elena Leyva-López, Andreas E. Voloudakis, Eduardo R. Bejarano, Jesús Méndez-Lozano

**Affiliations:** 1Consejo Nacional de Ciencia y Tecnología (CONACYT), Instituto Politécnico Nacional, CIIDIR-Unidad Sinaloa, Departamento de Biotecnología Agrícola, Guasave, Sinaloa 81101, Mexico; 2Instituto Politécnico Nacional, CIIDIR-Unidad Sinaloa, Departamento de Biotecnología Agrícola, Guasave, Sinaloa 81101, Mexico; 3Laboratory of Plant Breeding and Biometry, Agricultural University of Athens, 75 Iera Odos, Athens 11855, Greece; 4Instituto de Hortofruticultura Subtropical y Mediterránea “La Mayora”, Universidad de Málaga-Consejo Superior de Investigaciones Científicas (IHSM-UMA-CSIC), Universidad de Málaga, Campus Teatinos, 29071 Málaga, Spain

**Keywords:** geminivirus, high-throughput sequencing, non-cultivated plants, viral biodiversity, agro-ecological interface

## Abstract

Plant DNA viruses of the genus *Begomovirus* have been documented as the most genetically diverse in the family *Geminiviridae* and present a serious threat for global horticultural production, especially considering climate change. It is important to characterize naturally existing begomoviruses, since viral genetic diversity in non-cultivated plants could lead to future disease epidemics in crops. In this study, high-throughput sequencing (HTS) was employed to determine viral diversity of samples collected in a survey performed during 2012–2016 in seven states of Northern-Pacific Mexico, areas of diverse climatic conditions where different vegetable crops are subject to intensive farming. In total, 132 plant species, belonging to 34 families, were identified and sampled in the natural ecosystems surrounding cultivated areas (agro-ecological interface). HTS analysis and subsequent *de novo* assembly revealed a number of geminivirus-related DNA signatures with 80 to 100% DNA similarity with begomoviral sequences present in the genome databank. The analysis revealed DNA signatures corresponding to 52 crop-infecting and 35 non-cultivated-infecting geminiviruses that, interestingly, were present in different plant species. Such an analysis deepens our knowledge of geminiviral diversity and could help detecting emerging viruses affecting crops in different agro-climatic regions.

## 1. Introduction

Agroecosystems are used for the production of food, feed, fuel, fiber, and other harvestable goods providing human support and health [1]. Mexico has 196,437,500 ha, of which approximately 13% correspond to agricultural land. In 2016, 21.9 million ha were cultivated, with an agricultural production of 26,032 million tons with a value of 26,760 million dollars, which allowed the country to be ranked eleventh in the world´s crop production.

Plant diseases caused by begomoviruses and RNA viruses have been the main concern in Mexican horticulture throughout the years, having a significant negative impact on the crop production of tomato, pepper, bean, pumpkin, melon, soybean, tomatillo, tobacco, watermelon, and cotton [2,3,4,5,6,7]. Begomoviruses (family *Geminiviridae*) are characterized by their geminate particles that encapsidate a circular single-stranded (ss) DNA genome (monopartite or bipartite) of about 2.8 kb in size. They are transmitted via whitefly (*Bemisia tabaci*), infecting a large number of plant species worldwide and causing serious crop losses, and constitute a major global threat [8]. Important diseases caused by geminiviruses include maize streak disease [9], cassava mosaic disease [10], cotton leaf curl disease [11], and tomato leaf curl disease [12]. In Mexico, the first report of a disease caused in tomato by geminiviruses came from the Sinaloa state in 1970, and was later designated as *Chino del tomato virus* (CdTV) [13]. However, the first serious disease was reported in pepper crops and was named “rizado amarillo”, identified as the coinfection with *Pepper huasteco yellow vein virus* (PHYVV) and *Pepper golden mosaic virus* (PepGMV) [14]. Recently, the introduction of *Tomato yellow leaf curl virus* (TYLCV) in Sinaloa has dealt a devastating impact on tomato production. In different agro-climatic regions of Mexico, begomovirus-associated diseases are commonly caused by mixed infections with diverse begomoviruses, and affect tomato and pepper crops [5,15,16]. Coinfections with begomoviruses adapted to non-cultivated plants together with crop-adapted begomoviruses have been reported in soybean, tobacco, and pepper plants in the Sinaloa, Chiapas, and Jalisco states of Mexico [17,18,19]. Geminiviruses exhibit high mutation rates and recombination frequencies, both within and between species, resulting in rapid adaptive evolution. Several reports have indicated the emergence of recombinant species of geminiviruses [20,21,22,23]. For example, *Tomato yellow leaf curl Malaga virus* (TYLMaV) is a recombinant of *Tomato yellow leaf curl Sardinia virus* (TYLCSV) and *Tomato yellow leaf curl virus* mild strain (TYLCV) which, unlike its “parental” genomes, has the ability to infect common bean and wild *Solanum nigrum* [24]. More importantly, TYLMaV accumulated to the same levels in susceptible and resistant tomato, indicating that the recombination event and subsequent selection led to the generation of a resistance-breaking isolate [25].

It has been proposed that global warming will influence the epidemiology of virus-related plant diseases, mainly due to alterations in the distribution of insect vectors and in their host range [26,27]. Emerging diseases caused by geminiviruses have arisen more frequently in recent years, which emergence could be associated with climate change [28]. The viral quasispecies (non-identical but related genomes) present in a host plant could be generated by recombinant genomes providing an improved fitness potential to the viruses that may subsequently initiate an infection in new host species, or cause more severe disease symptoms in an established host by overcoming plant resistance. The generation of quasispecies depends on the host-virus interaction, the environmental conditions, and the cultivation practices [9,29,30,31]. 

New mutant or recombinant viral genomes that have arisen in non-cultivated plants could achieve successful infection of plant hosts to which they have newly adapted and cause significant crop damage, in a process subject to effective transmission between plant species. Vector-transmission is an evolutionary barrier for plant viruses to expand their host range. Alterations in vector-dependent transmission, most likely through mutations in the viral coat protein sequences, could increase the risk of emergence of a plant pathogen in a given crop. Begomoviruses are transmitted by the whitefly *Bemisia tabaci,* while aphid vectoring was recently confirmed for the *Capulovirus Alfalfa leaf curl virus* (ALCV) [32,33]. Vector metagenomics could contribute to surveying the diversity of geminiviruses present in their insect vectors and to defining the corresponding coat protein sequences. Although coat proteins have a structural role for the viral capsid, CP gene sequences are known to diverge [34] due to a high mutation rate. In addition, it is possible that such CP mutations could facilitate new means of transmission, such as seed transmission. Therefore, it is important to study the complexity and heterogeneity of geminiviral quasipecies found in wild reservoir hosts, and understand the factors behind the observed genetic variation since such knowledge will be extremely useful to develop resistance strategies (e.g., RNAi-based approaches) and, as a consequence, to prevent crop losses.

High-throughput sequencing (HTS) has provided the means to detect known viruses as well as to identify novel viruses in plants [35,36,37,38,39]. As a result, sensitive and accurate diagnosis of viral infection has been achieved, rendering this method extremely useful for quarantine purposes. The development of bioinformatic tools and the design of various pipelines have contributed significantly towards a deep analysis of the vast amount of HTS data produced [40].

The current next generation sequence (NGS) technologies have a couple of drawbacks: firstly, the contigs/singletons need to be annotated *de novo* via short read assembly, a process that may create chimeras deriving from the different genomes present in a sample, and secondly, the accurate differentiation of sequences; thus, confirmation is required by cloning followed by Sanger sequencing. The hope is that the latest single-molecule NGS technologies (3rd generation), where long reads are obtained, could address both of the above-mentioned issues, especially when their sequence reading error rates drop significantly compared to the levels of the previous NGS technologies. The resolution of the metagenome of a given sample could identify genetic variations in a viral population, providing the necessary input to study viral genome evolution, and determine which environmental factors might influence the generation of novel plant pathogens from previously benign viruses.

It is a well accepted notion that non-cultivated (wild) host plants play a key role in the generation of viral genetic variation, maintaining a certain degree of sequence heterogeneity convenient for viral adaptation in nature, without altering essential consensus sequences. Only recently, metagenomics studies on non-cultivated plant species have attracted the attention of plant researchers [38,41]. Mexico is considered to be one of the most megadiverse countries worldwide [42] and, despite the fact that some non-cultivated plant species have been reported as geminivirus reservoirs [16,43], to date the knowledge of geminivirus distribution in Mexican natural ecosystems is still limited. To this purpose, we aimed to determine the genetic diversity of begomoviruses in non-cultivated species that are present in the vicinity of cultivated crops (designated as the agro-ecological interface) in Northern-Pacific Mexico. In the present study, rolling circle amplification (RCA) was applied to DNA samples from wild plant species obtained during a five-year survey, and several begomoviral DNA signatures and begomoviral sequences were identified. Our results support the presence of a niche for begomoviral evolution in the neighborhood of important cultivation areas in Northern-Pacific Mexico.

## 2. Materials and Methods

### 2.1. Plant Sample Collection

A total of 422 non-cultivated plants (both symptomatic and asymptomatic) located in the area between crops and wild vegetation zones (agro-ecological interface), in seven states in the northern-pacific region of Mexico during the period 2012–2016, were collected, GPS-documented, photographed, and identified to the species level. Thus, 132 species of plants belonging to 34 families were identified. The sampling regions were grouped as follows: (1) Baja California (BC), (2) Sonora (SO), (3) Sinaloa (SI), (4) Colima-Nayarit (CN), and (5) Coahuila-Durango (CD) states of Mexico. Samples were placed on ice and brought to the laboratory and stored at −80 °C until processed. Plants were collected in non-protected areas; additionally, an herbarium was established with most of the plant specimens.

### 2.2. DNA Isolation, RCA, and Library Construction

Total DNA was extracted from individual plants using the CTAB method [44], and the isolated DNA, upon spectrophotometric estimation of its concentration, was used as template for PCR-based *Begomovirus* detection using degenerate universal primers (Appendix A). For each sampling region, total DNA from *Begomovirus* PCR-positive plants, belonging to the same plant species, was mixed in equimolar concentration. Following this procedure, 100 ng of each DNA mixture was used for circular DNA-molecule enrichment by rolling circle amplification (RCA) using the illustra TempliPhi DNA Amplification Kit (GE Healthcare, Milwaukee, MA, USA), following the manufacturer´s instructions. Then, all RCA products obtained per sampling region were pooled in equimolar concentrations, and cleaned using phenol:chloroform:isoamyl alcohol (25:24:1)/potassium acetate (5 M) and 100 ethanol % precipitation (1/10 *v*/*v*, 1/2 *v*/*v*, respectively). DNA integrity was analyzed by agarose gel electrophoresis, and the cleaned RCA mixtures were used for NGS library construction that was sequenced by a commercial facility (LANGEBIO, Irapuato, GTO, MX) using Illumina Nextera XT paired end 2 × 150 bp protocol on a MiSeq 500. The same procedure was followed for each sampling region to obtain one library per region (for a total of five libraries).

### 2.3. Metagenomic Analysis of Geminivirus-Related Signatures

Reads obtained from each library were trimmed employing the trimmomatic tool [45] (parameters (TRAILING: 30, HEADCROP:5) followed by quality check analysis by FASTQC (https://www.bioinformatics.babraham.ac.uk). Each library was filtered for human, bacteria, plant, and eukaryotic viruses reads using the ViromeScan pipeline [40] in order to obtain the geminivirus-related reads. All filtered libraries were subjected to de novo assembly using SPAdes [46]; both contigs (≥78 bp) and unassembled reads were compared against the GeneBank non-redundant database using BLASTn hosted in the Galaxy server [47]. Geminivirus-related signatures were sorted by contig length and analyzed manually. Contigs obtained in the present study are available at: https://www.dropbox.com/sh/ha6pkzls9217dhf/AAADNUa0TfYj3EZ8bb315cSga?dl=0.

### 2.4. Begomovirus Full-Length Genome Amplification, Cloning, and Sequence Analysis

Full-length geminiviral genomes were obtained following a previously described protocol [48]. In brief, the total DNA (100 ng) from selected non-cultivated plants was used as a template for viral circular DNA genome enrichment by RCA, as mentioned above. To obtain the viral monomeric full-length genomes, the RCA products were digested with a selected a single-site restriction enzyme (BamHI, EcoRI, XbaI, or XhoI, depending on the virus under analysis). The expected linearized geminivirus full-length genomes (~2.7 kb) were recovered from 1% ultrapure agarose gels using PureLink Quick Gel Extraction Kit (Thermo Fisher Scientific, USA) according to the manufacturer´s instructions. The fragments were ligated into linearized pGreen 0029 plasmid [49] that was digested with the corresponding restriction enzymes. The resulting recombinant plasmids were transformed in *E. coli* DH5α, and positive clones were subjected to Sanger genome walking sequencing. Genome assemblies were obtained using SeqMan (DNASTARInc, Madison, WI, USA) and SnapGene (GLS Biotech LLC, Chicago, IL, USA) software. All pairwise comparisons were performed using the MUSCLE algorithm implemented in Mega 7 [50] and maximum likelihood phylogenetic tree(s) were constructed on both begomoviral DNA components, with a 1000 bootstrap on both components to assess branch support. To analyze the nucleotide and amino acid identity, open reading frames (ORFs) were separated and individually compared with highest match homologous genome of each virus obtained from NCBI databank, using the ClustalW algorithm implemented in Mega 7.

## 3. Results and Discussion

It is an accepted notion that global warming will have an impact on global food security. In particular, crop yields are predicted to significantly decrease considering the “worst” CO_2_-emission scenario (A1FI) put forward by the Intergovernmental Panel on Climate Change [51]. Plant pathogens will have varying responses to climate change and plant-pathogen warfare is expected to be altered [52], imposing negative, neutral or positive effects on yields depending on the host-pathogen-environment tripartite interaction (known as the ‘disease triangle’). Disease pattern changes are anticipated due to alterations in the host range of plant pathogens, especially in rapidly evolving pathogens, while disease severity will be influenced by increased CO_2_, heavy rains, increased humidity, drought, and warmer winter temperatures [53]. Studies towards the understanding of the existing genetic diversity of plant viruses occurring in the agro-ecological interface, the generation of new genomes with features advantageous to the pathogen, and the relationship with their corresponding vectors will contribute significantly to humankind’s preparation to adapt to climate change and to sustain future food production. In this study, high-throughput sequencing (HTS) was employed to determine begomoviral diversity in plant samples collected in Northern-Pacific Mexico.

### 3.1. Non-Cultivated Plants from Northern-Pacific Mexico Region as a Reservoir of Begomoviruses

Plant viruses have generally been studied as disease-causing infectious agents that have a negative impact on their hosts [54]. Considering the wide diversity of geminiviruses, non-cultivated plants may serve as reservoirs for known, agriculturally relevant viruses: such weed-hosted viral species hold great potential to initiate the future emergence of new diseases in cultivated plants [55]. To determine begomoviral diversity in Northern-Pacific Mexico, a survey was performed in the agro-ecological interface in the vicinity of crop sites during 2012–2016. Sampling areas were divided in five regions: Baja California, Sonora, Sinaloa, Colima-Nayarit, and Coahulia-Durango, with three, four, seven, two, and four sampling points, respectively (Figure 1a). A total of 422 non-cultivated plants (both symptomatic and asymptomatic), belonging to 34 families and 132 species were identified (Appendix A). Among the different families identified in each region, the most commonly distributed were the *Astaraceae*, *Solanaceae*, *Malvaceae*, and *Fabaceae*. It is worth mentioning that those families were found to be the most predominant in natural ecosystems considering 200 plant families described in Mexico [42,56]. Using degenerate universal primers based on the DNA A genome of the genus *Begomovirus*, we were able to amplify the expected DNA fragment of 950–1100 bp in 252 out of the 422 (60%) tested individual plant specimens, indicating that begomoviruses were present in 29 plant families collected from all five regions sampled (Table 1). These results suggest that a number of non-cultivated plants widely distributed in Northern-Pacific Mexico represent a reservoir for begomoviruses.

### 3.2. Metagenomics Study Reveals a Number of Geminviruses from Non-Cultivated Plants

Following the pipeline described in the Materials and Methods section, NGS resulted in 16 to 215 million reads for the five libraries. After human, bacteria, plant, and eukaryotic virus sequence depletion, an NCBI-GenBank database search was performed to identify the most closely related geminivirus sequences, obtaining between 6000 and 4.6 million reads (Table 2). Subsequent annotation showed that more than 99% of the reads correspond to the genus *Begomovirus*, and the remaining 1% matches the *Geminiviridae* genera *Curtovirus*, *Becurtovirus*, *Turncurtovirus*, *Topocuvirus* and *Mastrevirus*. No sequences with homology to genera *Capulavirus*, *Eragrovirus*, and *Grablovirus* were identified in our study. The genus *Begomovirus* has been reported previously as the most widely distributed in Mexico [3,4,16,18,57,58,59]; in addition, sporadic reports on other genera like *Curtovirus* and *Grablovirus* were described [60,61]. Our data pointed out the abundance and importance of the genus *Begomovirus* in Mexico; however, follow up studies of the other genera are imperative.

The de novo assembly of geminivirus-related reads resulted in 24,289 geminivirus-related contigs ranging from 78 to 2858 bp in length (Table 2). The generated contigs were used to search the NCBI-GenBank database in order to identify the most closely related geminivirus exemplars at the species level (Appendix A). Table 3 shows all geminivirus-related signatures ≥300 bp and with at least 80% similarity at the nucleotide level with the best match, regardless whether DNA A or B viral components were detected. Similar findings were described in a metagenomics analysis in whiteflies, in which only one DNA component of a bipartite begomovirus was retrieved [62]. Additionally, the geminivirus-related signatures with 100–300 bp and/or below <80% similarity at the nucleotide level with the best match in NCBI gene sequences are listed in Appendix A. It is important to note that short geminivirus-related signatures could hinder the correct classification of a begomovirus species or strain; nonetheless, profiling the phylogenetic composition of the viral communities is pivotal as a significant part of the different plant-virus-environment interaction.

The analysis of the highest geminivirus-related signature sequences revealed a list of both bipartite and monopartite begomoviruses, including crop-adapted viruses in different plant families; 14 with bipartite genomes such as *Pepper huasteco yellow vein virus* (PHYVV-signature) present in four regions, *Pepper golden mosaic virus* (PepGMV-signature) present in four regions, *Pepper leafroll virus* (PepLRV-signature) present in one region, *Tomato chino la Paz virus* (ToChLPV-signature) present in two regions, *Tomato severe leaf curl virus* (ToSLCV-signature) present in two regions, *Tomato yellow spot virus* (ToYSV) present in three regions, *Potato yellow mosaic virus* (PYMV) present in one region, *Okra yellow mosaic mottle virus* (OYMMV-signature) present in four regions, *Cabbage leaf curl virus* (CabLCV-signature) present in two regions, *Bean calico mosaic virus* (BCaMV-signature) present in four regions, *Bean yellow mosaic Mexico virus* (BYMMV-signature) present in one region, *Vigna yellow mosaic virus* (ViYMV-signature) present in two regions, *Water melon chlorotic stunt virus* (WmCSV-signature) present in one region and *Squash leaf curl virus* (SLCV-signature) present in three regions; and five with monopartite genomes, three belonging to the genus *Begomovirus*, namely *Chilli leaf curl virus* (ChiLCV-signature) present in one region, and *Tomato yellow leaf curl virus* (TYLCV-signature) present in four regions, *Sweet potato leaf curl virus* (SPLCV-signature) present in one region; additionally, one belonging to the genus *Curtovirus*, namely *Beet curly top virus* (BCTV-signature) and another belonging to genus *Topocovirus*, namely *Tomato pseudo-curly top virus* (TPCTV), both present in one region (Table 3). Furthermore, nine non-cultivated plant-adapted viruses included only begomoviruses with bipartite genomes such as *Solanum mosaic Bolivia virus* (SoMBoV-signature), present in two regions, *Sida mosaic Sinaloa virus* (SiMSiV-signature) present in five regions, *Sida golden yellow spot virus* (SiGYSV-singnature) present in one region, *Malvastrum bright yellow mosaic virus* (MaBYMV-signature) present in three region, *Rhyncosia golden mosaic virus* (RhGMV-signature) present in five regions, and *Rhyncosia golden mosaic Sinaloa virus* (RhGMSV-signature) present in two regions, *Euphorbia mosaic virus* (EuMV-signature) present in three regions, *Euphorbia yellow mosaic virus* (EuYMV) present in one region and *Blechum leaf curl virus* (BlelCV-signature) present in one region, (Table 3). Interestingly, the observation of *Tomato pseudo-curly top virus* (TPCTV) in samples from Colima-Nayarit is the first report of the genus *Topocovirus* in Mexico. The list of geminiviruses are grouped as a potential of new viruses or strain of the best match virus, with molecular and biological validation being necessary.

The genus *Begomovirus* comprises the most common DNA viruses responsible for several virus-associated plant diseases in Mexico. Among them PHYVV, an endemic virus, and PepGMV have been documented as the most widespread and predominant in pepper crops [5,14,57,63]. It is noteworthy that the introduction of the promiscuous TYLCV in Yucatán and later in Sinaloa states, with a dramatic impact on crop yield, became the major concern for tomato crops in Northern Mexico [64]. Interestingly, TYLCV did not exclude the “native” viruses, and its co-infection with PHYVV or PepGMV caused severe disease in pepper crops [65]. However, the presence of such viruses in non-cultivated plants, as alternative hosts in nature, increases the chances of viral evolution through recombination events or other mechanisms, representing a latent threat. In fact, a new isolate of PHYVV described in pepper had significant sequence changes on its DNA B genome, which confers a modified host range since this isolate was able to infect tomato plants causing severe symptoms [5,66]. The other invasive virus which was introduced in Sonora state, apparently from the Middle East, was WmCSV [6], and it was detected in the present study along with SLCV. Our results suggest that WmCSV and SLCV are present in non-cultivated plants collected in Coahuila-Durango region (Table 3), which implies that WmCSV virus has the potential to spread in Mexico, and by adapting to new environments it has the potential to initiate an emerging disease in a new region. It is important to mention that both viruses could interact in mixed infection inducing more severe symptoms on crops as reported in Jordan [67]. The detection of ToChLPV, ToSLCV, OYMMV, BCaMV, and BYMMV, that were reported previously in Mexico, indicate that they still occupy an ecological niche and could be a potential source of viral disease. The identification of SiMSiV and RhGMV in all regions sampled is intriguing. Perhaps both of them represent viruses that are well adapted to different hosts and environments with a potential risk to evolve into an emerging disease. SiMSinV was initially reported in Sinaloa state associated to *Sida rombifolia* [17] without a negative impact on crops described to date. On the other hand, RhGMV was previously reported as the cause of disease in tobacco and soybean [4,18]. It is well known that viruses adapted to non-cultivated plants do not normally induce disease symptoms in their hosts. Nonetheless, SiMSiV and RhGMV induce symptoms in their corresponding first reported hosts (*Sida rombifolia* and *Rhyncosia minima,* respectively), with different wild species (acting as reservoirs) being suspected as the origin of the inoculum. Moreover, EuMV and EuYMV were reported previously in *Euphorbia heterophylla* in Mexico and Brazil [68,69]; whereas BlelCV, is a novel virus recently described in Chiapas state [70]. To the best of our knowledge, the ChiLCV, PepLRV, ToYSV, PYMV, CabLCV, SPLCV, MaBYMV, and ViYMV geminivirus-related signatures have not been previously described and/or associated to disease in Mexico and represent potential strains and/or novel viruses whose biological role needs to be determined in the immediate future. It is worth mentioning that viruses like ChilCV, PepLRV, and ToYSV are already associated with pepper, bean, and tomato diseases in Pakistan, Peru, Ecuador and Brazil [71,72,73].

### 3.3. Molecular Validation of the Predominant Begomoviruses Identified by HTS

The ecology of begomoviruses identified by HTS studies in non-cultivated plants requires more effort in order to understand the contribution of these plants for disease development [74,75,76]. Non-cultivated plants could be a source of viral inoculum to cultivated plants [77,78,79,80,81,82,83] and could contribute to viral evolution. Here, we described some non-cultivated plants at the agro-ecological interface carrying geminivirus-signatures (Table 3 and Appendix A). The information acquired represents important progress towards elucidating the above-mentioned issues, but more work is needed for the validation of the described identities.

According to our survey, plants belonging to the *Fabaceae*, *Malvaceae*, and *Solanaceae* families are the most widely distributed in all sampled areas. The HTS analysis revealed that signatures corresponding to TYLCV, SiMSiV, and RhGMV were present in all five NGS libraries, whereas the RhGMSV-signature was detected only in two out of five NGS libraries, suggesting that these species of begomovirus are predominant. Initially, the presence of these four viruses was confirmed by using sequence-specific primers for each viral species (Appendix A), testing an individual plant of the corresponding family for viral presence. To obtain the full-length viral genome of detected viruses, total DNA of *Nicotiana glauca* from Sinaloa (TYLCV PCR-positive); of *Sida acuta* from Colima (SiMSiV PCR-positive), and two *Rhynchosia minima* both from Sinaloa (PCR-positive for RhGMV/RhGMSV), were used for RCA amplification and cloning. Sequence analysis of obtained clones is summarized in Table 4 and Table 5.

Clone LV15-Ng-04 (Accession number: MK643155) from *N. glauca* was 2781 nt in length and showed high nucleotide homology (99.9%) to TYLCV (Accession number: EF523478.1). Clones LV15-Sa-03 and LV15-Sa-02 (Accession numbers: MK636866, and MK643154), from *S. acuta*, were 2611 nt and 2583 nt in length and showed nucleotide homology of 95.1 and 91.3% with DNA-A and DNA-B of SiMSiV (Accession numbers: DQ520944.1, and DQ356428.1, respectively). Clones LV17-Rm-02 LV17-Rm-06 (Accession numbers: MK634355, and MK634539), from *R. minima*, were 2605 nt and 2568 nt in length and showed nucleotide homology of 98.6 and 91% with DNA-A and DNA-B of RhGMV (Accession numbers: EU339939.1, and DQ356429.1, respectively). Finally, clones LV15-Rm-02 LV15-Rm-08 (Accession numbers: MK618662, and MK618663), from *R. minima*, were 2578 nt and 2525 nt in length and showed nucleotide homology of 91.9 and 85.9% with DNA-A and DNA-B of RhGMSV (Accession numbers: DQ406672.1, and DQ406673.1, respectively). For all viral genomes obtained, the predicted stem-loop region containing the sequence TAATATTAC, found in the common region of family *Geminiviridae*, was identified. For bipartite begomoviruses, high nucleotide homology of the common region (CR), of 98, 91, and 87% (DNA-A versus DNA-B) was observed for SiMSiV, RhGMV, and RhGMSV, respectively. Furthermore, the array of regulatory elements (iterons and TATA boxes) was conserved in all cases, suggesting that corresponding DNA-A and DNB-B are cognates. According to the present taxonomic classification of ICTV [84], for family *Geminiviridae*, the clones of TYLCV, SiMSiV, and RhGMV obtained in this study are classified as strains (≥94% DNA-A nucleotide homology), whereas the RhGMSV clones are classified as different isolates (≥91% DNA-A nucleotide homology).

Phylogenetic trees based on the nucleotide alignment with selected begomoviruses from the GenBank database, are shown in Figure 2. The results show that TYLCV isolate LV15-Ng-04 from *N. glauca* cluster together with different TYLCV isolates from different regions of the world, and segregate more closely to Mexican and Asian isolates (Israel and China) (Figure 2A). These data are in agreement with the TYLCV classification by geographic area, where Asian and American isolates are placed in Group I [85]. SiMSiV is a *Malvaceae*-infecting virus, whereas RhGMV and RhGMSV are *Fabaceae*-infecting viruses. Phylogenetic analysis showed that SiMSiV (DNA A and B) isolated from *S. acute* is closely related to an isolate of SiMSiV from Sinaloa state (Figure 2B,C). Similarly, isolates of RhGMV (DNA A and B) isolated from *R. minima* is clustered with isolates previously reported from soybean and weeds from Sinaloa state (Figure 2B,C). Altogether, the HTS analysis strongly suggests the existence of biologically active viruses in the agro-ecological interface with the potential of developing novel or emerging crop diseases. Finally, infectious clones of TYLCV, SiMSiV, RhGMV and RhGMSV were also obtained (to be described elsewhere).

### 3.4. Ecogenomic Analyis of Predominant Begomoviruses

The metagenomic approach carried out in the present work, allowed us to identify the geminiviral diversity present in different plant families and geographical regions; however, it is crucial from an ecological point of view to determine the occurrence and dynamics of the viral community in non-cultivated plants.

Ecogenomic analyses of begomovirus were accomplished by sequence-specific PCR detection including PHYVV, TYLCV, SiMSiV, and RhGMV/RhGMSV as the most widely distributed species in Northern-Pacific Mexico (Table 3, Appendix A). Thus, a total of 126 individual species sorted into the predominant plant families (*Fabaceae*, *Malvaceae*, and *Solanaceae*) were examined for the dynamics of the individual plant-virus infections in the five sampling regions (Table 6, and Appendix A). The analysis revealed that TYLCV was detected in 89 plants, emerging as the predominant virus, followed by SiMSiV, RhGMV/RhGMSV, and PHYVV with 63, 62, and 46 detections, respectively. Moreover, the number of single infections were lower, namely the observed corresponded to TYLCV (13.54%) followed by SiMSiV and RhGMV/RhGMSV (3.12 and 1.04%, respectively), suggesting that single infection is not common (Table 6, Figure 3). Interestingly, the dynamics of multiple (double, triple or quadruple) infections seem to be the rule in most of the plant species analyzed (Figure 3). The most common double viral infection detected was TYLCV-SiMSiV (14.58%), while TYLCV-SiMSiV-RhGMV/RhGMSV (11.45%) and TYLCV-PHYVV-RhGMV/RhGMSV (10.4%) were the most common triple infections. Furthermore, quadruple infections with TYLCV-PHYVV-SiMSiV-RhGMV/RhGMSV (31.25%) represented the highest viral complex found in different plant species and in the five agroecological regions included in these study (Table 6, and Figure 3).

TYLCV is the major viral concern for tomato production worldwide [78]. It was recently identified as a seed-transmitted virus [86] and is associated with crop diseases with other begomoviruses in both the Old and New World [87,88,89,90,91,92,93,94]. It has also been reported in many non-cultivated plant species worldwide including the families *Amaranthaceae, Asteraceae, Chenopodiaceae, Convolvulaceae, Euphorbiaceae, Fabaceae, Malvaceae*, and *Solanaceae* [95,96,97,98,99]. In the present work, TYLCV was detected in new host species of the *Fabaceae* family (*Crotalaria juncea*, *Lonchocarpus lanceolatus*, *Macroptilium atropurpureum*, *Melilotus indicus*, *Rhychosia precatoria*, *Rhynchosia minima and Senna uniflora*), the *Malvaceae* family (*Abutilon trisulcatum*, *Anoda pentaschista*, *Herissantia crispa*, *kosteletzkua depressa*, *Malvastrum coromandelianum*, *Malvella leprosa*, *Melochia piramydata*, *Sida acuta*, *Sida rombifolia*, *Sidastrum lodiegensis and Sphaeralcea angustifolia*), and the *Solanaceae* family (*Datura discolor*, *Nicotiana plumbanginifolia*, *Nicotiana glauca*, *Solanum trydynamum*, *Solanum verbacifolium*). PHYVV is an endemic virus of Mexico, described as a major concern for pepper production [5,14,63,65], additionally, it has been reported in several plant families [15,100]. Here, PHYVV was detected in new host species of the *Fabaceae* family (*C. juncea, R. precatoria, R. minima* and *S. uniflora*), the *Malvaceae* family (*A. trisulcatum*, *H. crispa*, *K. depressa*, *M. coromandelianum*, *M. piramydata*, *S. acuta*, *S. rombifolia*, *S. lodiegensis*), and the *Solanaceae* family (*Datura stramonium*, *N. glauca*, *N. plumbanginifolia*, *Solanum elaegnifolium*, *Solanum rostrarum* and *Solanum verbascifolium*). On the other hand, SiMSiV was reported infecting *Sida rombifolia* in Mexico; interestingly, this virus was detected not only in several other malvaceous plants (*A. palmeri, A. pentaschista, H. crispa, K. depressa, Malva parviflora, M. coronomandelianum*) but also in the *Fabaceae* family (*C. juncea, L. lanceolatus, M. atropurpureum, R. precatoria, R. minima, S. uniflora*) and the *Solanaceae* family (*N. plumbanginifolia, Datrua stramonium, N. glauca, Solanum elaeagnifolium, Solanum nigrescens and S. rostrarum*). Furthermore, RhGMV was first reported infecting *Rhynchosia minima* in Honduras [101] and after infecting tobacco and soybean crops in Chiapas and Sinaloa states from Mexico, respectively [4,18]. The data showed that RhGMV species (RhGMV and/or RhGMSV) were able to infect not only *Rhynchosia minima* but also other fabaceous plants (*C. juncea*, *Macroptilium atropurpureum*, *R. precatoria*, *S. uniflora*, *Meliotus indica*, and *L. lanceolatus*), and other hosts belonging to the *Malvaceae* (*A. palmeri*, *A. trisulcatun*, *H. crispa*, *K. depressa, M. parviflora*, *M. coromandelianum*, *M. piramydata*, *S. acuta*, *S. rombifolia*, and *S. lodiegensis*), and the *Solanaceae* (*D. stramonium, N. glauca*, *N. plumbanginifolia*, *S. nigrescens*, and *S. rostrarum*) families. This part of the study was oriented to individual plant samples, providing evidence about the ecology of the virus. The existence of several plant species harboring different viruses in multiple infections represents an important melting pot for the geminivirus to evolve in different directions triggered by vector and environmental factors [102]. Previous works have described extensively strains or new viruses in plants or whiteflies from different regions and some imply the host plant and biological properties [32,68,69,102]. Nonetheless, data from other studies were limited because the viral host was not determined [62,103]. The rate of discovery of new viral sequences is insufficient in terms of plant pathology [54,76], therefore an in-depth survey and biological determination is required to enhance our comprehension of the agroecological, environmental impact of viral evolution.

The evolution of plant viruses is a complex process involving multiple ecological and genetic factors resulting in host-pathogen co-evolution. Studies on viral diversity have been documented in a wide number of non-cultivated plants, with new entries described either as different strains of existing viruses or as new viruses [32,62,68,102,103,104,105,106]. However, it is noteworthy that viral species are often co-infecting the same plants with the possibility of genetic interaction, giving raise to inter- or intra-specific recombinant viruses resulting in more severe strains, as in the case of cassava mosaic diseases (CMD) [104,107] and tomato yellow curl diseases (TYLCD) [23,24,25,108]. Plant disease emergence requires that a virus from a reservoir host invades a new host resulting in new infection dynamics and viral adaptation [109]. In this sense, some geminiviruses acquired the ability to form a complex disease by assorting DNA A and B genomes, like ACMV and EACMV in Uganda [107] or those cases where the DNA A genome is similar to bipartite begomoviruses but the DNA B has not yet been described, such as TChLPV and ToSLCV in Mexico [7,58] or *Datura leaf curl virus* (DaLCV) in Sudan [110]. In the present work, the geminiviral diversity was described in different Mexican regions, showing that TYLCV, SiMSinV, PHYVV, RhGMV and RhGMSV are the predominant viruses. In addition, different multiviral complexes were detected in several plant species, highlighting the high frequency of mixed infections detected (Figure 3). However, a relevant issue is the wide host range observed for these viruses and that the genetic structure of the virome could be modified in an unpredictable manner. Moving forward, work is in progress aiming to determine whether new viruses or strains constitute a potential risk for Mexican agriculture.

## Figures and Tables

**Figure 1 viruses-11-00594-f001:**
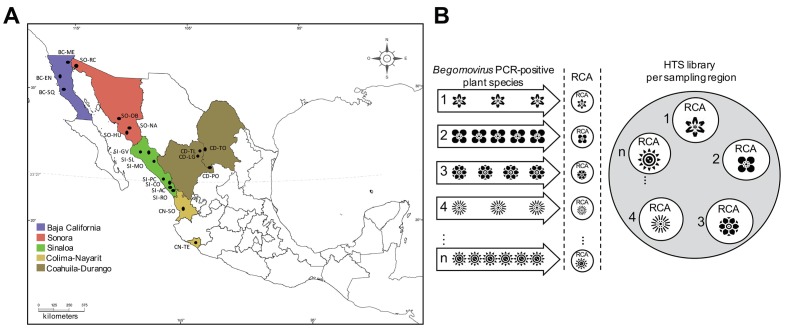
Northern-Pacific Mexico sampling areas and high-throughput sequencing (HTS) library construction. (**a**) Sampling areas were divided in five regions located in different biogeographic zones (according to the CONABIO classification): Baja California (BC), Sonora (SO), Sinaloa (SI), Colima-Nayarit (CN), and Coahulia-Durango (CD). BC-SQ: Baja California, San Quintin; BC-EN: Baja California, Ensenada; BC-ME: Baja California, Mexicali; SO-OB: Sonora, Obregon; SO-NA: Sonora, Navojoa; SO-HU: Sonora, Huatabampo; SO-RC: Sonora, Rio Colorado; SI-GV: Sinaloa, Guasave; SI-SL: Sinaloa, Sinaloa de Leyva; SI-MO: Sinaloa, Mocorito; SI-PC: Sinaloa, Playa Ceuta; SI-CO: Sinaloa, Concordia; SI-AC: Sinaloa, Agua Caliente; SI-RO: Sinaloa, El Rosario; CN-SO: Colima-Nayarit, Santa Maria del Oro; CN-TE: Colima-Nayarit, Tecoman; CD-TL: Coahuila-Durango, Tlahualilo; CD-LG: Coahuila-Durango, La Goma; CD-TO: Coahuila-Durango, Torreon; CD-PO: Coahuila-Durango, Poanas. Sampling areas are indicated in the map by colored squares. (**b**) Diagram of sample processing to obtain HTS libraries. For each sampling area, total DNA from *Begomovirus* PCR-positive plants belonging to the same species was pooled in equimolar concentrations. The resulting DNA mix was used as a template for rolling circle amplification (RCA)-mediated viral circular molecule enrichment. Finally, all resulting RCA products were pooled in equimolar concentrations and used for HTS library construction.

**Figure 2 viruses-11-00594-f002:**
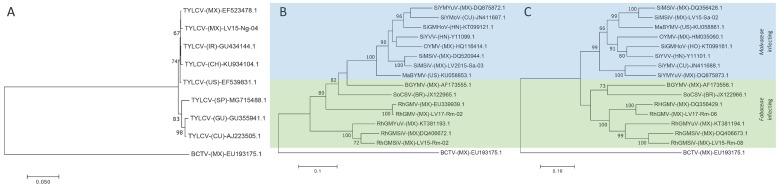
Phylogenetic trees based on multiple sequence alignment of complete monopartite (**A**) and bipartite begomovirus DNA-A (**B**), and DNA-B (**C**) with selected isolates obtained from NCBI. Trees were constructed by the maximum likelihood method with 1000 bootstrap replicates using MEGA7. Virus acronyms: *Bean golden yellow mosaic virus* (BGYMV), *Malvastrum bright yellow mosaic virus* (MaBYMV), *Okra yellow mosaic virus* (OYMV), *Rhynchosia golden mosaic Sinaloa virus* (RhGMSV), *Rhynchosia golden mosaic virus* (RhGMV), *Rhynchosia golden mosaic Yucatan virus* (RhGMYuV), *Sida golden mosaic Honduras virus* (SiGMHoV), *Sida mosaic Sinaloa virus* (SiMSiV), *Sida yellow mottle virus* (SiYMV), *Sida yellow mosaic Yucatan virus* (SiYMYuV), *Sida yellow vein virus* (SiYVV), *Soybean chlorotic spot virus* (SoCSV). Viral genomes Accession numbers are shown. Countries codes are as follow: Brazil (BR), Cuba (CU), Guatemala (GU), Ecuador (EC), Honduras (HN), Israel (IR), Mexico (MX), Puerto Rico (PR) and United States of America (US). As an out-group, *Beet curly top virus* sequence (BCTV) was used. *Malvaceae* and *Fabaceae* infecting virus are highlighted in blue and green, respectively.

**Figure 3 viruses-11-00594-f003:**
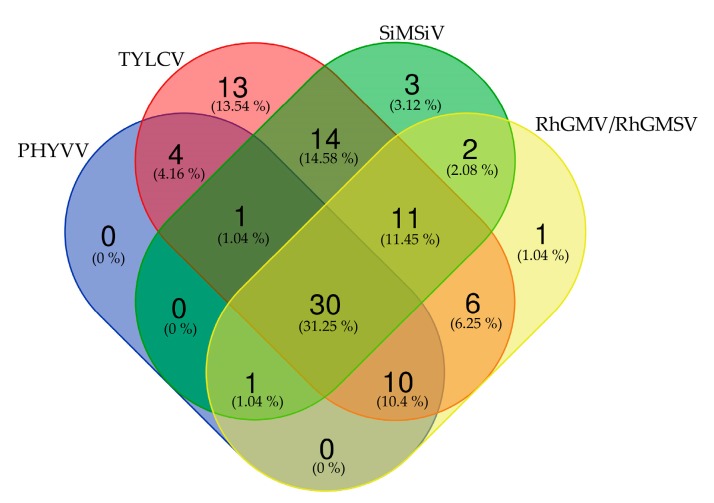
Venn diagram of predominant begomovirus detected. A total of 126 individual species were examined for the presence of predominant begomovirus in the five sampling regions. The number of begomovirus-positive plants (97) showing single, double, triple or quadruple infections is reported. The values are also reported in percentage in parenthesis. Virus acronyms: *Pepper huasteco yellow vein virus* (PHYVV), *Rhynchosia golden mosaic virus*, (RhGMV), *Rhynchosia golden mosaic Sinaloa virus* (RhGMSV), *Sida mosaic Sinaloa virus* (SiMSiV), *Tomato yellow leaf curl virus* (TYLCV).

**Table 1 viruses-11-00594-t001:** Begomovirus detection in plants collected in the agro-ecological interface from Northern-Pacific regions of Mexico. Total DNA from each individual specimen was used as a template for PCR-mediated begomovirus detection.

Plant Family ^1^	Sampling Region ^2^ Begomovirus PCR-Positive/Total Plants Collected				
	BC	SO	SI	CN	CD
*Amaranthaceae*		4/5	1/4	1/4	5/6
*Apiaceae*		1/1			
*Asteraceae*	4/10	8/16	8/22	3/4	0/15
*Boraginaceae*	0/1	4/5			
*Brassicaceae*	1/2	0/2	1/1		
*Caesalpiniaceae*			1/1		
*Capparaceae*			1/2		
*Chenopodiaceae*	4/4	6/10			
*Convolvulaceae*	0/2	2/2	3/4	1/4	
*Cucurbitaceae*	0/1		2/6	1/3	
*Euphorbiaceae*		2/3	5/15	0/2	
*Fabaceae*		1/2	49/72	0/1	
*Hydrophyllaceae*		1/1		0/1	
*Malvaceae*	5/6	14/18	21/33	9/13	9/9
*Menispermaceae*			1/1		
*Nyctaginaceae*			2/2	4/5	1/1
*Onagraceae*		1/2			
*Papaveraceae*		1/1			
*Pedaliaceae*			1/1		
*Polygonaceae*	1/2	3/5			
*Portulacaceae*			3/3	1/2	
*Primulaceae*	1/1				
*Rhamnaceae*		1/1	1/1		
*Rubiaceae*				2/2	
*Sapindaceae*			1/1		
*Solanaceae*	1/2	11/14	18/22	0/2	14/14
*Sterculiaceae*			1/2		
*Verbenaceae*			1/2	2/2	
*Vitaceae*			1/1	0/1	
Total positives	17	60	122	24	29

^1^ Plant families PCR-negative for begomovirus detection: *Apocynaceae*, *Asclepiadiaceae*, *Bignoniaceae*, *Commelinaceae* and *Malpiguiaceae*. ^2^ Plant specimens belonging to 34 families were collected from five regions: Baja California (BC), Sonora (SO), Sinaloa (SI), Colima-Nayarit (CN), Coahuila-Durango (CD).

**Table 2 viruses-11-00594-t002:** Next generation sequence (NGS) data summary of reads and contigs mapping to Geminivirus genomes.

Library Name	Total Reads	Total Geminivirus-Related Reads	Number of Geminivirus-Related Contigs	Smallest/Largest Geminivirus-Related Contig ^1^
Baja California	16,056,866	6156	92	78/2437
Sonora	30,440,802	23,546	195	78/2293
Sinaloa	215,007,456	4,685,423	15,465	78/2723
Colima-Nayarit	33,159,620	2,475,219	8368	78/2775
Coahuila-Durango	70,782,034	349,763	169	78/2858
Total	365,446,778	7,540,107	24,289	78/2858

^1^ bp: Base pairs.

**Table 3 viruses-11-00594-t003:** Begomovirus signatures obtained by *de novo* assembly from the metagenomics study in plants collected in the agro-ecological interface from five regions in Northern-Pacific Mexico. Geminivirus-related reads for each NGS library were used for de novo assembly and generation of signatures.

Host Adapted	Virus Acronym ^2^	Plant Family of First Detection	Geminivirus-Signatures of DNA-A/DNA-B ^1^ per Region				
			Baja California	Sonora	Sinaloa	Colima-Nayarit	Coahuila-Durango
Crops	PHYVV	*Solanaceae*	ND ^3^	98.5/96.5 LN848858.1/LN848912.1 1462/2124	99.6/100 LN848873.1/KP890828.1 251/594	100/99.6 X70418.1/X70419.1 (583/1826)	96.9/98.5 LN848872.1/LN848922.1 955/1742
	PepGMV		ND/98.4 ND/AY928515.1 ND/524	ND	98.1/95.9 U57457.1/AY928515.1 1115/2388	88.2/84.3 AY905553.1/LN848829.1 136/147	99.4/95.9 LN848772.1/LN848841.1 1120/1562
	PepLRV		ND	88/ND KC769819.1/ND 458/ND	ND	ND	ND
	ChiLCV		ND	ND	81.2/NA ^4^ JN555601.1/NA 559/NA	ND	ND
	TYLCV		98.4/NA JQ354991.1/NA 131/NA	99.5/NA KU836749.1/NA 2540/NA	99.5/NA FJ012359.1/NA 1247/NA	99.7/NA EF523478.1/NA 1524/NA	99.4/NA FJ012358.1/NA 2048/NA
	ToChLPV		ND	89.1/NA AY339618.1/NA 120/NA	ND	81.9/NA HM459852.1/NA 337/NA	ND
	ToSLCV		ND	87.2/NA DQ347946.1/NA 359/NA	ND	99.5/NA KC479066.1/NA 411/NA	ND
	TPCTV		ND	ND	ND	82.3/NA X84735.1/NA 385/NA	ND
	ToYSV		ND	84.2/ND DQ336350.1/ND 470/ND	95.4/ND KJ742419.1/ND 155/ND	84.9/ND KX348173.1/ND 192/ND	ND
	PYMV		ND	ND	ND	78.8/ND FR851299.1/ND 321/ND	ND
	OYMMV	*Malvaceae*	ND/90.3 ND/GU972604.1 ND/2354	ND	93.6/96.4 GU990612.1/JX219471.1 174/226	98.9/94.9 GU990614.1/JX219471.1 1455/336	ND/94.9 ND/JX219471.1 ND/236
	CabLCV	*Brassicaceae*	ND	ND	97.4/ND AJ228570.1/ND 119/ND	84.2/82.8 MH359394.1/DQ178613.1 1645/157	ND
	BCaMV	*Fabaceae*	ND	ND/95 ND/AF110190.1 ND/2576	97.2/96.7 AF110189.1/AF110190.1 2058/1296	92.3/88.9 AF110189.1/AF110190.1 353/135	97.9/82.9 AF110189.1/AF110190.1 1005/587
	BYMMV		ND	85.3/ND FJ944023.1/ND 677/ND	ND	ND	ND
	ViYMV		ND	ND	86.6/86.7 KC430936.1/KC430937.1 758/369	89.6/86 KC430936.1/KC430937.1 242/115	ND
	WmCSV	*Cucurbitaceae*	ND	ND	ND	ND	100/100 KY124280.1/KY124281.1 239/1025
	SLCV		ND	94.2/ND KM595165.1/ND 104/ND	ND	80.6/83 KM595183.1/DQ285017.1 155/124	79.8/95.3 KM595165.1/M38182.1 188/1649
	SPLCV	*Convolvulaceae*	ND	ND	92.4/NA KX611145.1/NA 1818/NA	80/NA KJ013582.1/NA 261/NA	ND
	BCTV	*Amaranthaceae*	99.8/NA JX487184.1/NA 508/NA	ND	ND	ND	ND
Non-cultivated plants	SoMBoV	*Solanaceae*	ND	ND/84.7 ND/HM585436.1 ND/518	ND	ND/82.3 ND/HM585436.1 ND/655	ND
	SiMSiV	*Malvaceae*	96.3/ND DQ520944.1/ND 854/ND	95.8/96.7 DQ520944.1/DQ356428.1 2581/1582	96.9/90.2 DQ520944.1/DQ356428.1 1003/2085	95.6/87.7 DQ520944.1/DQ356428.1 1584/245	94.2/98.9 DQ520944.1/DQ356428.1 572/289
	SiGYSV		ND	84.6/ND KX348185.1/ND 637/ND	ND	ND	ND
	MaBYMV		96.9/ND KU058856.1/ND 1037/ND	ND	97.2/84.9 KU058865.1/KU058860.1 403/153	ND	94.8/94.5 KU058853.1/KU058859.1 1822/1282
	RhGMV	*Fabaceae*	95/90 EU339939.1/EU339937.1 1049/536	88.9/ND EU021216.1/ND 253/ND	98.9/95.7 EU339939.1/EU339937.1 2086/675	92.2/85.4 EU339938.1/DQ356429.1 155/240	96.2/82.6 AF408199.1/EU339937.1 264/543
	RhGMSV		ND	ND	93.4/96.2 DQ406672.1/DQ406673.1 1754/1794	91.2/89.2 DQ406672.1/DQ406673.1 727/353	ND
	EuMV	*Euphorbiaceae*	ND	86.8/ND JN368145.1/ND 678/ND	ND	87.9/86.5 DQ318937.1/DQ520942.1 158/104	ND/93.1 ND/HQ185235.1 ND/249
	EuYMV		ND	ND	ND	91.3/80.1 KY559516.1/KY559581.1 138/342	ND
	BleICV	*Acanthaceae*	ND	ND	ND/79 ND/JX827488.1 ND/783	ND	ND

^1^ Best match in %, accession numbers and contig length aligned is shown. Contig alignments of ≥300 bp in length were selected regardless whether one or both viral components (DNA A and B) were detected. Contigs alignments of <300 bp are also reported if at least one signature of ≥300 bp for the corresponding virus was detected. ^2^ Virus acronyms: *Monopartite Geminiviruses: Beet curly top virus* (BCTV), *Chilli leaf curl virus* (ChiLCV), *Sweet potato leaf curl virus (SPLCV), Tomato pseudo-curly top virus* (TPCTV), *Tomato yellow leaf curl virus* (TYLCV); *Bipartite Geminiviruses: Bean calico mosaic virus* (BCaMV), *Blechum interveinal chlorosis virus* (BleICV), *Bean yellow mosaic Mexico virus* (BYMMV), *Cabbage leaf curl virus* (CabLCV), *Euphorbia mosaic virus* (EuMV), *Euphorbia yellow mosaic virus* (EuYMV), *Malvastrum bright yellow mosaic virus* (MaBYMV), *Okra yellow mosaic Mexico virus* (OYMMV), *Pepper golden mosaic virus* (PepGMV), *Pepper huasteco yellow vein virus* (PHYVV), *Pepper leafroll virus* (PepLRV), *Potato yellow mosaic virus* (PYMV), *Rhynchosia golden mosaic Sinaloa virus* (RhGMSV), *Rhynchosia golden mosaic virus* (RhGMV), *Sida golden yellow vein virus* (SiGYVV), *Sida mosaic Sinaloa virus* (SiMSiV), *Squash leaf curl virus* (SLCV), *Solanum mosaic Bolivia virus* (SoMBoV), *Tomato chino la Paz virus* (ToChLPV), *Tomato severe leaf curl virus* (ToSLCV), *Tomato yellow spot virus* (ToYSV), *Vigna yellow mosaic virus* (ViYMV), *Watermelon chlorotic stunt virus* (WmCSV). ^3^ND: No detected. ^4^ NA: Not applicable.

**Table 4 viruses-11-00594-t004:** Nucleotide and amino acid sequence identities (%) between DNA-A genome of Begomovirus isolates identified in the present study with best match sequences available in the database.

Clon Code	Length (bp)	Accession No.	Virus Acronym ^1^	Reference Genome	Complete Genome	Virus Gene ^2^											
						CP		V2		Rep		TrAp		REn		C4	
					N ^3^	n	a ^4^	n	a	n	a	n	a	n	a	n	a
LV15-Ng-04	2781	MK643155	TYLCV	EF523478.1	99.9	99.6	100	99.7	99.1	99.9	100	99.8	100	99.5	98.5	100	100
LV15-Sa-03	2611	MK636866	SiMSiV	DQ520944.1	95.1	96.3	98.4	NA ^5^	NA	94.9	95.8	96.5	93.7	95.6	93.2	94.8	98.4
LV17-Rm-02	2605	MK634355	RhGMV	EU339939.1	98.6	98.8	100	NA	NA	98.5	98.9	98.5	97.1	98.9	97.7	99.2	97.7
LV15-RM-02	2578	MK618662	RhGMSV	DQ406672.1	91.9	91	95.2	NA	NA	92.9	92.3	96.9	95.3	95.3	93.9	92	86.6

^1^ TYLCV: *Tomato yellow leaf curl virus*, SiMSiV: *Sida mosaic Sinaloa virus*, RhGMV: *Rhynchosia golden mosaic virus*, RhGMSV: *Rhynchosia golden mosaic Sinaloa virus*. ^2^ CP (V1): Coat protein, V2: Precoat protein, Rep (C1): Replication associated protein, TrAp (C2): Transcriptional activator protein, REn (C3): Replication enhancer protein, C4: C4 protein. ^3^ n: Nucleotide homologies in %. ^4^ a: Aminoacidic homologies in %. ^5^ NA: Not applicable.

**Table 5 viruses-11-00594-t005:** Nucleotide and amino acid sequence identities (%) between DNA-B genome of Begomovirus isolates identified in the present study and best match sequences available in the database.

Clon Code	Length (bp)	Accession No.	Virus Acronym ^1^	Reference Genome	Complete Genome	MP ^1^		NSP ^2^	
					n ^3^	n	a ^4^	N	a
LV15-Sa-02	2583	MK643154	SiMSiV	DQ356428.1	91.3	92.7	95.6	90.3	92.2
LV17-Rm-06	2568	MK634539	RhGMV	DQ356429.1	91	94.8	99.3	91.7	91.6
LV15-Rm-08	2525	MK618663	RhGMSV	DQ406673.1	85.9	90.4	98.3	83.3	86.7

^1^ MP: Movement protein. ^2^ NSP: Nuclear shuttle protein. ^3^ n: Nucleotide homologies in %. ^4^ a: Aminoacidic homologies in %.

**Table 6 viruses-11-00594-t006:** Specific PCR detection of begomovirus in individual non-cultivated plants collected from different counties of the five Northern Pacific-regions of Mexico included in this study.

Sampling Area	Plant Family	Plant Species	Collection Year	Virus ^1^ Specific PCR-Positive Samples				Negative Samples	Total Samples
				PHYVV	TYLCV	SiMSiV	RhGMV/RhGMSV		
**BAJA CALIFORNIA**									
Ensenada	*Malvaceae*	*Malva parviflora*	2015	ND ^2^	1	ND	ND	0	1
	*Solanaceae*	*Nicotiana glauca*	2015	ND	1	1	1	0	1
San Quintin	*Malvaceae*	*Malva parviflora*	2015	ND	3	1	ND	1	4
**SONORA**									
Huatabampo	*Malvaceae*	*Abutilon palmeri*	2015	ND	1	1	1	1	2
		*Abutilon trisulcatun*	2015	ND	1	1	ND	0	1
		*Anoda pedunculosa*	2015	ND	ND	ND	ND	1	1
	*Solanaceae*	*Datura stramonium*	2015	1	1	1	1	0	1
		*Nicotiana glauca*	2015	ND	1	1	ND	1	2
		*Nicotiana plumbanginifolia*	2015	ND	1	1	ND	0	2
		*Solanum. spp*	2015	1	1	ND	1	0	1
		*Solanum nigrescens*	2015	ND	ND	1	ND	0	1
		*Solanum. spp*	2015	1	1	1	1	0	1
		*Solanum verbascifolium*	2015	1	1	ND	ND	0	1
Navojoa	*Fabaceae*	*Meliotus indica*	2015	ND	1	ND	1	0	1
	*Malvaceae*	*Malva parviflora*	2015	ND	2	3	1	2	5
		*Malvella leprosa*	2015	ND	1	1	ND	0	1
Obregón	*Malvaceae*	*Abutilon palmeri*	2015	ND	ND	ND	ND	1	1
		*Sida rombifolia*	2015	1	1	1	1	0	1
	*Solanaceae*	*Nicotiana glauca*	2015	1	1	1	1	0	1
		*Nicotina plumbanginifolia*	2015	1	1	1	1	0	1
Río Colorado	*Malvaceae*	*Malva parviflora*	2015	ND	1	1	ND	0	1
**SINALOA**									
Agua caliente	*Fabaceae*	*Rhynchosia minima*	2016	4	4	1	3	1	5
Concordia	*Malvaceae*	*Anoda pentaschista*	2012	ND	1	ND	ND	0	1
Guasave	*Fabaceae*	*Crotalaria juncea*	2016	1	3	2	3	0	3
		*Lonchocarpus lanceolatus*	2012	1	1	1	1	0	1
		*Melilotus indicus*	2015	ND	ND	ND	ND	1	1
			2016	1	1	ND	1	0	1
	*Malvaceae*	*Abutilon palmeri*	2012	ND	1	ND	ND	0	1
		*Abutilon trisulcatun*	2014	ND	1	ND	1	2	3
			2012	1	1	ND	1	0	1
		*Herissantia crispa*	2012	ND	1	ND	1	0	1
		*Kosteletzkya depressa*	2012	2	2	2	2	0	2
		*Melochia piramydata*	2014	4	4	4	4	0	4
	*Solanaceae*	*Datura reburra*	2012	ND	1	ND	ND	0	1
		*Datura stramonium*	2012	ND	ND	ND	ND	1	1
			2014	3	3	ND	3	1	4
		*Nicotiana glauca*	2012	ND	1	ND	1	1	2
		*Solanum americanum*	2012	ND	ND	ND	ND	1	1
		*Solanum nigrescens*	2012	ND	1	ND	1	0	1
Mocorito	*Malvaceae*	*Abutilon trisulcatun*	2012	1	1	ND	ND	1	2
		*Sidastrum lodiegensis*	2012	1	1	ND	1	0	1
	*Solanaceae*	*Datura discolor*	2012	ND	ND	ND	ND	2	2
		*Solanum tridynamum*	2012	ND	ND	ND	ND	1	1
Playa Ceuta	*Fabaceae*	*Rhynchosia minima*	2016	2	2	2	2	0	2
Rosario	*Fabaceae*	*Macroptilium atropurpureum*	2016	2	3	4	4	0	4
		*Rhynchosia precatoria*	2016	2	2	2	2	0	2
		*Rhynchosia minima*	2016	3	4	4	4	0	4
		*Senna uniflora*	2016	1	ND	1	1	0	1
			2014	1	1	1	1	0	1
	*Malvaceae*	*Abutilon trisulcatun*	2014	ND	1	ND	1	0	1
		*Herissantia crispa*	2014	1	1	1	1	0	1
		*Melochia piramydata*	2014	ND	ND	ND	ND	1	1
		*Sida acuta*	2014	2	2	2	2	0	2
	*Solanaceae*	*Physalis acutifolia*	2014	ND	ND	ND	ND	1	1
Sinaloa		*Datura inoxia*	2012	ND	1	ND	ND	0	1
		Solanum tridynamum	2012	ND	2	ND	ND	1	3
**COLIMA/NAYARIT**									
Tecomán	*Malvaceae*	*Herissantia crispa*	2014	1	5	5	5	0	5
		*Malvastrum coromandelianum*	2014	1	1	1	2	1	3
		*Sida acuta*	2014	ND	ND	+	ND		
**COAHUILA/DURANGO**									
La Goma	*Malvaceae*	*Sida acuta*	2015	ND	ND	ND	ND	1	1
		*Sida rombifolia*	2015	ND	2	1	ND	0	2
	*Solanaceae*	*Datura stramonium*	2015	ND	1	1	ND	0	1
Poanas		*Solanum elaeagnifolium*	2016	ND	ND	ND	ND	2	2
		*Solanum rostrarum*	2016	2	3	3	3	0	3
Tlahualilo	*Malvaceae*	*Sida rombifolia*	2015	ND	ND	1	1	2	3
	*Solanaceae*	*Solanum elaeagnifolium*	2015	1	1	1	ND	0	1
Torreón	*Malvaceae*	*Sphaeralcea angustifolia*	2015	ND	2	ND	ND	1	3
	*Solanaceae*	*Datura stramonium*	2015	ND	1	1	ND	0	1
		*Nicotiana glauca*	2015	1	2	1	ND	1	3
		*Solanum elaeagnifolium*	2015	ND	3	3	ND	0	3
**Total**				**46**	**89**	**63**	**62**	**30**	**126**

^1^ Virus acronyms: PHYVV, Pepper huasteco yellow vein virus; TYLCV, Tomato yellow leaf virus; SiMSiV, Sida mosaic Sinaloa virus; RhGMV, Rhynchosia golden mosaic virus; RhGMSV, Rhynchosia golden mosaic Sinaloa virus. ^2^ ND: Not detected.

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
