# Peer review of "High-Throughput Sequencing Reveals Differential Begomovirus Species Diversity in Non-Cultivated Plants in Northern-Pacific Mexico"

_viruses, 2019, doi:10.3390/v11070594_

Round 1
Reviewer 1 Report
The manuscript of Rodriguez-Negrete et al. describes geminivirus survey in the natural ecosystems in Mexico. The authors collected about 400 non-cultivated plants in 5 regions of Mexico. Total DNA was extracted from each individual plant. About half of them were positive in PCR reactions when using degenerated primers for begomovirus detection, specific for DNA-A component. DNAs from PCR-positive plants belonging to the same species, from a given region, were mixed and circular DNAs present in the samples was RCA-enriched. In the next step, all RCA products from plants collected from a given region were mixed to constitute one library per region, which was subjected to high throughput sequencing (HTS). Five DNA libraries were constructed this way. DNA reads obtained for each library were de novo assembled into contigs, the contigs were blasted against NCBI GenBank database. Table 3 presents results of alignments of contigs of 500-2700 bp in length with >90 % nucleotide identity; 13 known geminiviruses were identified. Contigs of 200-500 bp in length aligned to additional 11 geminiviruse sequences (<90 % nucleotide identity), table S2.
To prove the real presence of viruses detected by HTS, DNAs extracted from four individual plants (if I understood correctly) were used to clone a full-length geminivirus.
Line 323-326: « Total DNA from Nicotiana glauca from Sinaloa, TYLCV PCR-positive; Sida acuta from Colima, SiMSiV PCR-positive, and two Rhynchosia minima both from Sinaloa, PCR-positive for RhGMV and RhGMSV, respectively, was used for RCA-guided circular viral genomes enrichment. »
Sequences of these 4 cloned viruses were compared with known geminiviruses, phylogenetic tree is present in Figure 2.
These are interesting data, but the manuscript requires major revisions. The biological material and HTS sequencing data are very valuable but not deeply exploited.
1. I have the impression that HTS data were not fully analysed.
1.1. How could it be that for several viruses DNA-A was not identified, table 3: case of OYMMV and PepGMV for BC rejoin, BCaMV for Sonora, PHYVV for Sinaloa, WmCSV and SLCV for CD region? May be the original HTS sequences should be mapped to the sequences of viruses identified by Blast comparisons of NODEs (contigs) in table 3 and S2.
1.2. I just opened one folder from contigs obtained in the present
study that are available in:
https://www.dropbox.com/sh/ha6pkzls9217dhf/AAADNUa0TfYj3EZ8bb315cSga?dl=0. For CN region, the NODE2 of 2651 in length gives 84 % similarity with cabbage leaf curl virus, but this virus is not mentioned at all in the MS. I did not check other folders.
2. Line 321-323: « Initially, the presence of those viruses was confirmed by using specific primers for each viral species, in which an individual plant of the corresponding family was tested for virus presence (data not shown) ». I think these data should be shown. You have DNA extracts from individual plants. 252 plants were positive in PCR using degenerated primers for geminiviruses. This is not a big deal to analyse them with primers specific for the viruses that you have identified by HTS. The results may reveal the real co-infection of one plant by several viruses, as you indicate in line 275, but I do not have ‘Morales-Aguilar 2019 in press’ data).
3. How many clones were sequenced to give the consensus sequence of your 4 cloned viruses. What is variability between sequences of cloned viruses and the corresponding contigs obtained by HTS?
4. To complete your study on geminivirus variability in Mexico, I suggest to clone at least one of them from different plant species and from different regions. PHYVV could be the best candidate, as it seems to be different in 4 regions, according to your table 3. Back-to-back two primer pairs for PCR amplifications may give quick results.
5. Please provide all primer sequences, the reference to which you address the reader, Mauricio-Castillo et al 2007, addresses again to the previous papers.
6. Table 3. Give the length of contigs showing similarity to the indicated geminiviruses
7. Table 4 and 5, give the length of cloned DNAs, the information will be better readable than in the text, line 353-363.
7. Table S1. Provide sampling date and information if a plant was positive in PCR.
8. The manuscript is poorly written and requires careful revision by an English speaking person.
Author Response
Response to the Reviewer’s comments
Reviewer #1:
The manuscript of Rodriguez-Negrete et al. describes geminivirus survey in the natural ecosystems in Mexico. The authors collected about 400 non-cultivated plants in 5 regions of Mexico. Total DNA was extracted from each individual plant. About half of them were positive in PCR reactions when using degenerated primers for begomovirus detection, specific for DNA-A component. DNAs from PCR-positive plants belonging to the same species, from a given region, were mixed and circular DNAs present in the samples was RCA-enriched. In the next step, all RCA products from plants collected from a given region were mixed to constitute one library per region, which was subjected to high throughput sequencing (HTS). Five DNA libraries were constructed this way. DNA reads obtained for each library were de novo assembled into contigs, the contigs were blasted against NCBI GenBank database. Table 3 presents results of alignments of contigs of 300-2700 bp in length with >80 % nucleotide identity; 13 known geminiviruses were identified. Contigs of 200-500 bp in length aligned to additional 11 geminiviruse sequences (<90 % nucleotide identity), table S2.
To prove the real presence of viruses detected by HTS, DNAs extracted from four individual plants (if I understood correctly) were used to clone a full-length geminivirus.
Line 323-326: « Total DNA from Nicotiana glauca from Sinaloa, TYLCV PCR-positive; Sida acuta from Colima, SiMSiV PCR-positive, and two Rhynchosia minima both from Sinaloa, PCR-positive for RhGMV and RhGMSV, respectively, was used for RCA-guided circular viral genomes enrichment.»
Sequences of these 4 cloned viruses were compared with known geminiviruses, phylogenetic tree is present in Figure 2.
These are interesting data, but the manuscript requires major revisions. The biological material and HTS sequencing data are very valuable but not deeply exploited.
1. I have the impression that HTS data were not fully analysed.
R1. Initially we decided to consider only identities of ³90% and counting length starting from 200 or 500bp. Following the reviewer´s comment and based in the feasibility and confident analysis of ViromeScan pipline (Ramppeli et al 2016) the parameters were modified, including shorter fragments for the overall analysis (Modified Table 3 and Supplementary Table S2). Table 3 shows the geminivirus-related signatures ³300 bp and ³80% nucleotide homology against the best match regardless whether DNA A or B viral components were detected. Additionally, the geminivirus-related signatures with size of 100-300 bp and/or<80% nucleotide homology against the best match in NCBI gene sequences are listed in Supplementary Table S2.
1.1. How could it be that for several viruses DNA-A was not identified, table 3: case of OYMMV and PepGMV for BC rejoin, BCaMV for Sonora, PHYVV for Sinaloa, WmCSV and SLCV for CD region? May be the original HTS sequences should be mapped to the sequences of viruses identified by Blast comparisons of NODEs (contigs) in table 3 and S2.
R1.1. The observation is correct. However, even after revisiting our NGS data for some geminiviruses as PepGMV and OYMV, DNA A genome was still not detected in Baja California, such outcome could be due to the number of reads obtained in this library. Please note that similar results were reported by Rosario et al (2015), were only one component (DNA A) was retrieved on whiteflies studies.
For WmCSV and SLCV where DNA B (1025 and 1649 bp conting length respectively) was first described, the DNA A was excluded during the initially submitted text due their small sizes; in the revised text they are included (239 and 188 bp conting length respectively).
The decision of not perform map analysis is due the fact that DNA samples from different plant species were mixed for RCA amplification. If mapping was employed this could increase the possibility for assembling chimeric viruses.
1.2. I just opened one folder from contigs obtained in the present
study that are available in:
https://www.dropbox.com/sh/ha6pkzls9217dhf/AAADNUa0TfYj3EZ8bb315cSga?dl=0. For CN region, the NODE2 of 2651 in length gives 84 % similarity with cabbage leaf curl virus, but this virus is not mentioned at all in the MS. I did not check other folders.
R1.2. The initial idea was to considered only countings with homology ³90 %. Upon our new analysis, we present the new results in the modified Table 3.
2. Line 321-323: « Initially, the presence of those viruses was confirmed by using specific primers for each viral species, in which an individual plant of the corresponding family was tested for virus presence (data not shown) ». I think these data should be shown. You have DNA extracts from individual plants. 252 plants were positive in PCR using degenerated primers for geminiviruses. This is not a big deal to analyse them with primers specific for the viruses that you have identified by HTS. The results may reveal the real co-infection of one plant by several viruses, as you indicate in line 275, but I do not have ‘Morales-Aguilar 2019 in press’ data).
R2. Specific PCR detection was performed in 126 individual plant species belonging to the three main plant families (Solanaceae, Fabaceae, Malvaceae). The results are now presented in the revised text in new Table 6 and Supplementary Figure S2.
Morales-Agular et al 2019, is now published on line and is included in the references of the revised manuscript.
3. How many clones were sequenced to give the consensus sequence of your 4 cloned viruses. What is variability between sequences of cloned viruses and the corresponding contigs obtained by HTS?
R3. One clone for each virus was fully sequenced in order to confirm the identity and validate our analysis.
The geminivirus-signatures were variable in size; for this reason is not possible the verify the variability between them
4. To complete your study on geminivirus variability in Mexico, I suggest to clone at least one of them from different plant species and from different regions. PHYVV could be the best candidate, as it seems to be different in 4 regions, according to your table 3. Back-to-back two primer pairs for PCR amplifications may give quick results.
R4. We agree with the reviewer´s comment that the examination of individual viruses could provide valuable insights of geminivirus variability in Mexico. However, this was not the scope of the present manuscript. Due the relevance of the issue we plan to address in a near future research effort.
5. Please provide all primer sequences, the reference to which you address the reader, Mauricio-Castillo et al 2007, addresses again to the previous papers.
R5. A supplementary table S4 of primers sequences is now is included in the revised manuscript.
6. Table 3. Give the length of contigs showing similarity to the indicated geminiviruses
R6. The modification suggested was performed in the modified Table 3 and Supplementary Table S2.
7. Table 4 and 5, give the length of cloned DNAs, the information will be better readable than in the text, line 353-363.
R7. The modification was performed as suggested in Tables 4 and 5.
7. Table S1. Provide sampling date and information if a plant was positive in PCR.
R7. Table S1 was modified in the revised text as suggested by the reviewer.
8. The manuscript is poorly written and requires careful revision by an English speaking person.
R8. We believed that the manuscript text was improved in its revised form

Reviewer 2 Report
Lot of information regarding the methodology used are redundant in the text and need be to amended:
1: Paragraph 3.1 described the methodology rather than Results and Discussion. Therefore, it should be re-written to highlight results only (as from lines 203-208).
2: Again, Paragraph 3.2 (lines 229-233) is redundant. Re.phrase to highlight only results.
3: Lines 241-242: Explain why there should be an homology with Capulavirus, Eragrovirus, etc.
4: Lines 326-330: Redundant paragraph. it describes the methodology again. Re-phrase
5:Correct digit errors in Table 4.
Author Response
Response to the Reviewer’s comments
Reviewer #2:
Lot of information regarding the methodology used are redundant in the text and need be to amended:
1: Paragraph 3.1 described the methodology rather than Results and Discussion. Therefore, it should be re-written to highlight results only (as from lines 203-208).
R1. Was modified as indicated.
2: Again, Paragraph 3.2 (lines 229-233) is redundant. Re.phrase to highlight only results.
R2. Was modified as indicated.
3: Lines 241-242: Explain why there should be an homology with Capulavirus, Eragrovirus, etc.
R3. After quality control and trimming process, the metagenomic reads (60-150 nt length) were taxonomically classified using the ViromeScan pipeline containing 1,646 viral DNA sequences (human and insect) and 3,207 sequences of Geminiviridae family (containing the nine reported genera). The results showed that 99.9% of geminivirus-related reads share high homology with Begomovius genera, and the remaining 0.1% with other five genera (Becurtovirus, Curtovirus, Turncurtovirus, Topocuvirus, and Mastrevirus). Metagenomic reads-related to the remaining three genera (Capulavirus, Eragovirus, and Grablovirus), were not detected due these genera are not present in the samples of our study. However, we cannot rule out the possibility of presence of these viruses in low concentrations, or their presence in other regions of Mexico.
4: Lines 326-330: Redundant paragraph. it describes the methodology again. Re-phrase
R4. The text was modified as suggested.
5: Correct digit errors in Table 4.
R5. Correction was done as suggested.

Round 2
Reviewer 1 Report
The authors followed recommendations, they presented additional data, answered my questions and changed accordingly the text to make several points clearer. The revised version is significantly improved.
Some sentences still need to be changed to improve meaning or English:
Line 41 "Plant diseases caused by begomovirus among other RNA viruses…" - Begomoviruses are DNA viruses, so they are not among other RNA viruses. Do you mean "Plant diseases caused by begomoviruses and RNA viruses…"?
Line 58 - use "high mutation rate" instead of "high mutations rates"
Line 74 - use "..could arise in non-cultivated plants and upon transmission…" instead of "than upon transmission"
Line 105-106 - "Only recently, metagenomics studies on non-cultivated plant species have recently attracted.." delete one "recently" word.
Line 200 - "indicating" instead of "indicated"
Line 222 - "negative" instead of "negatives"
Line 231 - "to the genus" instead of "to thegenus"
Line 236 - "Our data pointed out on the abundance "
Line 237 - do you mean "follow-up studies of other genera become imperative"?
Line 253 - use "sequences" instead of "sequence"
Line 255 - use "14 with" instead of "with 14"
Line 264 - "in three regions"
Line 265 - should not it be "three" instead of "two"?
Line 270 - I think should be modified like following "...nine non-cultivated plant-adapted viruses included only begomoviruses with…"
Line 279 - delete comma after "Colima-Nayarit
Line 304 - delete "t" at the end of the line
Line 306 - "plant-adapted viruses normally do not induce.."
Line 314 - I do not understand the phrase "novel viruses in which the biological role waits to be determined", re-phrase please
Line 318-319 "The ecological role of begomoviruses…. needs more efforts in order to understand…"
The role does not need efforts, we need more efforts. Please re-phrase
Line 325 - "based on our survey"
Line 331-332 "To characterize at the molecular level and confirm the biological nature of detected viruses" I do not understand what do you mean under "the biological nature". Do you mean what first you amplified short fragment using virus-specific primers and when you wanted to confirm the presence of the full-length viral DNA?
Line 413 - analysis
Line 438 - Not detected
Line 482 - "previous works substantially describe….. from different regions" - plural because you site 4 papers.
Line 483 - "Nonetheless, data from other studies were limited" would be more clear than "nonetheless, others were limited"
Probably, publishing editor may find other grammar errors to correct.
I suggest to publish the manuscript after these corrections.

Author Response
Response to the Reviewer 1
Dear Reviewer 1
We would like to thanks all your comments and suggestion in order to improve the quality of our manscript.
The manuscript was revised according the comments and corrections using the “Track Changes” function in Microsoft Word as indicated.
Some sentences still need to be changed to improve meaning or English:
Line 41 "Plant diseases caused by begomovirus among other RNA viruses…" - Begomoviruses are DNA viruses, so they are not among other RNA viruses. Do you mean "Plant diseases caused by begomoviruses and RNA viruses…"?
R. The text was modified as suggested: Plant diseases caused by begomoviruses and RNA viruses…
Line 58 - use "high mutation rate" instead of "high mutations rates"
R. The text was modified as suggested (Line 59).
Line 74 - use "..could arise in non-cultivated plants and upon transmission…" instead of "than upon transmission"
R. The text was modified as suggested (Line 75).
Line 105-106 - "Only recently, metagenomics studies on non-cultivated plant species have recently attracted.." delete one "recently" word.
R. The text was modified as suggested (line 110).
Line 200 - "indicating" instead of "indicated"
R. The text was modified as suggested (line 207).
Line 222 - "negative" instead of "negatives"
R. The text was modified as suggested (line 230).
Line 231 - "to the genus" instead of "to thegenus"
R. The text was modified as suggested (line 239).
Line 236 - "Our data pointed out on the abundance "
R. The text was modified as suggested (line 244).
Line 237 - do you mean "follow-up studies of other genera become imperative"?
R. The text was modified as suggested (line 246).
Line 253 - use "sequences" instead of "sequence"
R. The text was modified as suggested (line 264).
Line 255 - use "14 with" instead of "with 14"
R. The text was modified as suggested (line 266).
Line 264 - "in three regions"
R. The text was modified as suggested (line 275)
Line 265 - should not it be "three" instead of "two"?
R. The text was modified, is correct the observation “three” instead of “two” (line 276).
Line 270 - I think should be modified like following "...nine non-cultivated plant-adapted viruses included only begomoviruses with…"
R. The text was modified as suggested (lines 281-282).
Line 279 - delete comma after "Colima-Nayarit
R. The text was modified as suggested (line 290)
Line 304 - delete "t" at the end of the line
R. The text was modified as suggested (line 317)
Line 306 - "plant-adapted viruses normally do not induce.."
R. The text was modified as suggested (line 319)
Line 314 - I do not understand the phrase "novel viruses in which the biological role waits to be determined", re-phrase please
R. The text was modified as suggested (line 328). Re-phrase as follow:
“novel viruses in which the biological role needs to be determined in the immediate future”.
Line 318-319 "The ecological role of begomoviruses…. needs more efforts in order to understand…" The role does not need efforts, we need more efforts. Please re-phrase
R. The text was modified as suggested (line 331). Re-phrase as follow:
“The ecology of begomoviruses….needs more efforts”
Line 325 - "based on our survey"
R. The text was modified as suggested (line 338).
Line 331-332 "To characterize at the molecular level and confirm the biological nature of detected viruses" I do not understand what do you mean under "the biological nature". Do you mean what first you amplified short fragment using virus-specific primers and when you wanted to confirm the presence of the full-length viral DNA?
R. The text was modified as suggested (lines 345-348
“To obtain the full-length viral genome of detected viruses, total DNA of Nicotiana glauca from Sinaloa (TYLCV PCR-positive); of Sida acuta from Colima (SiMSiV PCR-positive), and two Rhynchosia minima both from Sinaloa (PCR-positive for RhGMV/RhGMSV), was used for RCA amplification and cloning.”
Line 413 – analysis
R. The text was modified as suggested (line 439)
Line 438 - Not detected
R. The text was modified as suggested (line 460)
Line 482 - "previous works substantially describe….. from different regions" - plural because you site 4 papers.
R. The text was modified as suggested (line 503)
Line 483 - "Nonetheless, data from other studies were limited" would be more clear than "nonetheless, others were limited"
R. The text was modified as suggested (line 505)
Probably, publishing editor may find other grammar errors to correct.
I suggest to publish the manuscript after these corrections.
